# Marine Biotechnology: Challenges and Development Market Trends for the Enhancement of Biotic Resources in Industrial Pharmaceutical and Food Applications. A Statistical Analysis of Scientific Literature and Business Models

**DOI:** 10.3390/md19020061

**Published:** 2021-01-26

**Authors:** Sara Daniotti, Ilaria Re

**Affiliations:** Consorzio Italbiotec, 20138 Milan, Italy; ilaria.re@italbiotec.it

**Keywords:** marine biotechnology, drugs, food, market trend, Horizon 2020, pharmaceutical applications, TRL, Blue Growth

## Abstract

Biotechnology is an essential tool for the sustainable exploitation of marine resources, although the full development of their potential is complicated by a series of cognitive and technological limitations. Thanks to an innovative systematic approach that combines the meta-analysis of 620 articles produced worldwide with 29 high TRL (Technology Readiness Level) European funded projects, the study provides an assessment of the growth prospects of blue biotechnologies, with a focus on pharmaceutical and food applications, and the most promising technologies to overcome the main challenges in the commercialization of marine products. The results show a positive development trend, with publications more than doubled from 2010 (36) to 2019 (70). Biochemical and molecular characterization, with 150 studies, is the most widely used technology. However, the emerging technologies in basic research are omics technologies, pharmacological analysis and bioinformatics, which have doubled the number of publications in the last five years. On the other hand, technologies for optimizing the conditions of cultivation, harvesting and extraction are central to most business models with immediate commercial exploitation (65% of high-TRL selected projects), especially in food and nutraceutical applications. This research offers a starting point for future research to overcome all those obstacles that restrict the marketing of products derived from organisms.

## 1. Introduction

Through the analysis of the most recent scientific production and cases of industrial use, the study offers an overview of marine biotechnologies and the challenges associated with the exploitation of biotic resources for industrial pharmaceutical and food applications. The introduction sets out the potential of marine biotechnologies and the sources of bioactive compounds used to obtain products, goods, and services, focusing on the analysis of challenges associated with exploration of the marine environment, the sustainable production of substances with high added value, and the improvement of the competitiveness of marine bioproducts.

The methodological assumption underlying the choice of materials and the approach to systematic analysis lies in the contribution of scientific research to the definition of methods, technologies, and solutions aimed at overcoming the application limits of marine resources in the pharmaceutical and food sectors. The analysis of development trends, combined with the most innovative results obtained from research funded by the European Commission as part of the Horizon 2020 Program, enables us to assess the growth prospects of the blue biotechnology sector.

### 1.1. Marine Biotechnology: Definition and Industrial Applications in the Pharmaceutical and Food Sectors

With 70% of the Earth’s surface covered by oceans, it is estimated that the marine environment is home to over a million macroscopic species—algae, corals, sponges, mollusks, fish and mammals—not to mention approximately one billion species of microorganisms—viruses, bacteria, archaea, microalgae and fungi. Oceanic regions can also differ widely in terms of chemical and physical conditions such as temperature, salinity, availability of light and level of toxic compounds, which means that marine organisms have developed unique features that differ according to the environment. The enormous vastness of the seas and the variability of the species they host have led to a structural diversity of natural products, encouraging research into new bioactive substances extracted from these organisms, with extraordinary industrial application potential [1,2]. Algae, which include macroalgae, microalgae and cyanobacteria, are particularly promising because of the significant number of bioactive molecules they can generate, [3,4] along with microorganisms which, due to their ability to adapt to extreme conditions, are an invaluable source of enzymes and compounds with specific properties, for example, resistance to high temperatures or high salinity [5,6,7]. However, the oceans still remain largely unexplored, and little is known about the life they host. Scientists estimate that only 9% of marine species are known, with the remaining 91% yet to be classified [8]. Technological advances are enriching this knowledge, while also revealing how much remains to be discovered. The novelty of the sector is therefore further stimulating interest and research [9].

In this context, marine biotechnology—or blue biotechnology—is defined by the Organization for Economic Cooperation and Development (OECD) as the application of science and technology to living organisms from marine resources, as well as parts, products or models thereof, for the production of knowledge, goods and services. Therefore, the discipline is aimed at the development of products and tools related to marine bio-resources, which, in this sense, are considered in the dual role of source for the development of products with high added value, and targets of exploratory technologies for studying the marine environment [10].

The applications of this technology are numerous, ranging from the production of biofuels, particularly bioethanol through the fermentation of macro and microalgae [11], to the extraction of enzymes for the paper, textile and detergent industries, and laboratory applications.

Among the known examples, the enzyme Pfu polymerase, produced by an extremophilic marine microorganism—*Pyrococcus furiosus*—is commonly used for PCR where a high replication fidelity is required [7].

All those bioactive compounds obtained from marine organisms that can be used in the pharmaceutical, cosmetic and food industries are also extremely interesting and promising. Marine biotechnologies are, for example, widely used to extract minerals, fibers, and secondary metabolites such as lipids and carotenoids from macro-, micro-algae and cyanobacteria to be used as food supplements or nutraceutical additives; fish are also the most abundant, commercially used source for the extraction of omega-3 polyunsaturated fatty acids with proven cardioprotective and antioxidant properties [12]. These compounds also play a fundamental role in the cosmetic industry, by providing new active ingredients with antioxidant, moisturizing, anti-inflammatory and photo-protective properties to be added to creams and lotions [13].

However, only a limited number of medicinal products derived from marine organisms have been approved by regulatory authorities, since most of the products tested do not make it past the preclinical trials. There are, however, some examples of medicinal products that are commercially available and commonly used in medical practice, for example the analgesic Prialt^®^, the antihypertensive Lovaza^®^, and several anticancer agents such as Yondelis^®^ and Cytosar-U^®^ [14]. In 2020, the turnover of pharmaceutical products of marine origin in Europe, although limited in number, was almost 400 million dollars, with the highest market share value in the blue biotechnology sector. In terms of revenue generated, the food sector follows with 290 million dollars. Together with nutraceuticals, pharmaceutical products account for more than 60% of the European marine biotechnology market [15].

These examples demonstrate the remarkable achievements of marine biotechnology and highlight the feasibility of its application. They are therefore to be considered among those tools that, if appropriately managed and funded, will have the greatest impact on the economy, not only helping to develop the Blue Economy, but also to achieve a mature bioeconomy [10,16].

### 1.2. Main Marine Resources

The oceans are a source of organisms with huge diversity in terms of their features, physiology and, consequently, the potential secondary metabolites produced. Microorganisms represent the most promising resource of natural molecules because, unlike macroorganisms, they have the advantage to be sustainably cultivated on a large scale, at low cost [17].

Microalgae are among the most widely used organisms in this sector due to the high volume of compounds they can generate—vitamins, proteins with essential amino acids, polysaccharides, fatty acids, sterols, pigments, fibers, and enzymes—[4] whose quality in terms of chemical structure and activity is often better than their synthetic counterparts obtained in the laboratory [14]. Moreover, as photosynthetic organisms, they can be easily cultivated in photobioreactors or open ponds, exploiting solar energy and greenhouse gases present in the air to obtain abundant biomasses while helping to mitigate the concentration of air pollutants [3].

The microalgae market is mainly dominated by two species: *Chlorella* and *Spirulina.* The first is a green microalga belonging to the broad phylum of Chlorophyta, which includes both microalgae and macroalgae. It is widely used in cosmetics, food and pharmaceuticals for its anti-inflammatory, antimicrobial and anticancer properties [4]. On the other hand, *Spirulina*, characterized by a high protein content that makes it an excellent addition to foodstuffs, belongs to the phylum of Cyanobacteria, photosynthetic microorganisms also called blue-green algae. Due to the specific cellular mechanisms of these microalgae, they are readily adaptable to environmental conditions and able to grow rapidly [18].

Fungi and bacteria also possess unique characteristics that make them an excellent source of bioactive metabolites. They generally live in symbiosis with invertebrate organisms whose defense is strictly dependent on chemical compounds produced by these microorganisms; moreover, in response to the extreme nature of the environments in which they live, fungi and bacteria have developed specific properties that are reflected in the metabolites they produce [17].

Of particular interest are the actinobacteria, a large phylum of gram positives considered to be a mine of secondary metabolites. It is estimated that, up until now, approximately one third of the 3000 molecules with antibiotic activity isolated from microorganisms have been derived from these bacteria. Most of these bioactive substances have been isolated from the genus *Streptomyces*, which is extremely widespread in both marine and terrestrial environments [19].

There are also the proteobacteria, a variegated phylum of gram negatives that have aroused considerable interest in the scientific community due to the high variety of secondary metabolites they generate, with unusual structures that distinguish them from both their terrestrial counterparts and from actinobacteria. Relevant features of these substances include halogenation, sulfur-containing heterocycles, non-ribosomal peptides, and polyketides with a specific structure reflected in a greater range of activities, many of which are not found in other species [20].

Macroorganisms such as corals, sponges and other invertebrates, fish and sharks, are also a source of bioactive substances. However, technological limitations in their exploitation make their production unsustainable and it is often not possible to reach the quantities required on an industrial scale [21]. Macroalgae, as photosynthetic organisms, are among the most easily cultivated macroorganisms for use in food production as they are a source of metabolites with unique nutritional properties such as furanones, polyunsaturated fatty acids, pigments, phycocolloids and phlorotannins. Some varieties of red algae are also used for the production of agarose, which is used as a thickening and stabilizing agent in the food industry [12,14].

### 1.3. The Challenges Faced by Marine Biotechnology in Sustainable Industrial Development

In recent decades, numerous products derived from marine organisms have been placed on the market owing to innovative or established technologies that respond to the need to exploit the great potential that the marine environment offers, while safeguarding its fragile resources. Despite this progress, the marine biotechnology sector in the European and global markets is still marginal and not fully developed. For the purposes of this study, the main challenges to be addressed for this industry to expand and for more products to be marketed have been grouped into three categories: “discovery of new products”, “sustainable production”, “improvement of yield and product characteristics”.

#### 1.3.1. Discovery of New Products

Our knowledge of the marine environment is extremely limited, to the extent that it is estimated that over 90% of this biodiversity is completely unknown [8]. One of the main limitations in exploring the marine environment concerns the technical difficulty of accessing deep seas, which could be overcome by innovative technology developed outside the field of biology. For example, remotely controlled vehicles could facilitate the collection of samples from places that humans cannot reach; technology for automatic data collection would enable identifying areas with high biodiversity. [22].

Moreover, although many organisms are isolated from the marine environment, they are difficult to analyze and therefore taxonomic classification is complex, with potential errors that could compromise the entire process of drug discovery due to the impossibility of reproducing the isolation event and the subsequent identification of the bioactive compound [14]. Innovative analytical techniques, not only for the exploration and sampling of new organisms but also for the isolation, purification, characterization and analysis of the bioactivity of the compound, would therefore make it possible to overcome this limitation [23].

To address the high costs and long lead times of conventional screening programs, the pharmaceutical industry uses modern high-throughput screening methods to identify new pharmaceutical leads originating from the marine environment, automatically testing over 10,000 potential bioactive compounds per week, which greatly increases the success rate [24].

Combining these innovative screening techniques with bioinformatics is more successful than relying on experimental approaches alone when searching for new bioactive substances of marine origin. Recent advances in Information Technology have in fact led to the development of computational techniques that provide more efficient and more targeted research than the simple analysis of the genome or structure of the compound. They reveal the mechanism of action without the need for experiments and facilitate the optimization of pharmaceutical leads [25].

Also of recent interest are omics technologies, which play an increasingly significant part in determining the genetic capabilities of marine organisms. The sequencing of different microbial genomes has led to the discovery of a greater number of proteins per genome than those known, highlighting the existence of a still unexplored reserve of bioactive substances not expressed in the traditional culture conditions used in the laboratory.

Moreover, by applying metagenomic techniques to the samples collected and analyzing the genetic content of environmental samples, we can identify new bioactive substances produced by microorganisms found in the environment, hitherto unknown because they cannot be grown in the laboratory. By introducing their DNA isolated from the environment into appropriate hosts, DNA libraries are created; these can be easily screened for substances of interest. The main limitations of this promising technique are related to the impossibility of obtaining intact genes from the environment and the incompatibility of the expression elements used in libraries, obstacles that could be overcome by the most recent advances in synthetic biology [26,27]. The discovery of bioactive compounds with antibiotic activity such as violaceins, terragins and turbomycins through metagenomic techniques demonstrates their potential in exploiting non-cultivable microorganisms for drug discovery [28].

#### 1.3.2. Sustainable Production

The fragility of marine ecosystems must be taken into account when using marine organisms for the production of high value-added substances and biofuels. In fact, the quantities supplied directly by marine organisms do not support industrial requirements and the more limited ones of the drug discovery process. The direct collection of bioactive substances and other compounds of industrial interest is therefore almost never sustainable, especially since many of these species are in danger of extinction and their excessive exploitation could damage the delicate balance of the ecosystem.

Alternative solutions are therefore becoming necessary. Until now, conventional chemical and microbiological approaches have often involved the chemical synthesis or semi-synthesis of something similar to the natural product. However, these methods have a significant environmental impact, mainly due to the use of toxic solvents and large amounts of waste produced. Researchers and companies working in the field of marine biotechnology can therefore solve this limitation by developing more eco-friendly technologies [29]. The use of aquaculture and alternative methods for the cultivation of marine organisms such as corals and sponges undoubtedly constitutes a sustainable production method, since the use of toxic compounds is not necessary and the raw materials can be supplied without damaging fragile marine ecosystems [21,30].

A negative environmental impact of the production of bioactive compounds from marine organisms is also linked to the conventional downstream processes used in extracting the product: for example, the conventional method applied to microalgae not only requires a considerable amount of energy and produces organic waste, but it often has a prohibitive cost, which can amount to 80% of the cost of the entire process. The urgent need to develop alternative downstream processes can once again be solved by marine biotechnology [31].

The concept of sustainable production also includes the use of waste from the fishing industry to obtain substances with high added value. Indeed, waste from fish processing industries combined with by-catch generates a significant amount of waste, which exceeds dozens of millions of tons per year and raises several concerns about the environmental impact and the possibility of disposal. However, the extreme chemical richness of this waste means that it can be converted through biotechnological processes into products with high added value, such as protein hydrolysates with antioxidant properties or biomaterials with high market demand (e.g., collagen). This process therefore represents, on the one hand, a promising solution for the disposal of waste deriving from the fish supply chain and, on the other, an alternative method to the direct collection of bioactive substances [32,33].

Finally, reference must be made to the development of “green” enzymatic methods for the synthesis of products in order to reduce the use of toxic compounds associated with chemical synthesis. For example, nanoparticles which, due to their size, show unique properties for chemical, physical, and biomedical applications, have recently been developed. The use of algae or microorganisms for their preparation is a more sustainable alternative to the conventional method of nanoparticle synthesis, which is highly polluting due to the organic compounds used [34]. More sustainable conversion processes such as biocatalysis can derive benefits from numerous enzymes extracted from marine organisms that can adapt to extreme conditions in marine habitats. Many of these enzymes also possess stereochemical and catalytic properties that differ from their terrestrial counterparts. The best known examples are the esterase extracted from *Yarrowia lipolytica*, used in the racemic resolution of an ester to obtain a potent antibacterial agent, and a protease with broad substrate specificity that can be used as a generic catalyst for peptide cyclisation [35].

#### 1.3.3. Improving the Yield and Characteristics of the Product

The marketing of a consumer product is inevitably linked to its large-scale production. Many promising substances tested in the laboratory do not reach the market due to their uncompetitive production costs compared with the alternatives extracted from other organisms, from fossil-derived materials, or obtained synthetically.

In order to fulfil the high potential of marine resources, it is essential to develop processes that can supply a biomass compatible with market demand, with high yields and a resulting increase in competitiveness [36]. To this end, there is a growing development of new genetic and metabolic engineering techniques, with particular focus on microalgae [37,38]; optimization of the culture conditions of the organism not only affects the amount of biomass produced, but also the type and amount of secondary metabolites [39,40] and the extraction process [41,42]. Technological advances in bioreactor structures have also meant that volumes can be scaled up considerably, since they can mimic the marine environment in which organisms grow in order to maximize their productivity. Systems that can obtain the energy to function from the microbial metabolism itself, or in situ systems that use underwater modules deposited on the seabed to enable the sustainable exploitation of marine resources, are particularly innovative [43].

In some cases, the properties of the natural product must be improved to make it more effective in terms of activity—in the case of a medicinal product—or to give it features with more consumer appeal. Spirulina is a case in point: despite its high nutritional value and excellent therapeutic properties, the use of this microalgae as a food supplement is hindered by its unpleasant smell and the limited availability of proteins for human consumption. To solve this problem, ethanol-based fermentation and extraction processes have been developed. These can generate products with a smell and taste that are more desirable to the consumer, while also improving the availability of nutrients [44].

### 1.4. Purpose of the Publication

Marine resources represent a largely untapped resource that can be used with a limited carbon and environmental footprint to produce food, feed, and pharmaceuticals, addressing major societal challenges, including discovering new pharmaceutical products and the sustainable production of food to meet the exponential demographic growth.

Although marine biotechnology represents the most promising tool for the sustainable exploitation of these resources, many challenges prevent the commercialization of innovative marine-derived products since they do not meet the required volumetric production and economic competitiveness.

Analysis of technological advances and sustainable business models leading the market in the upcoming years are fundamental tools to identify the most promising methods, enhance blue biotechnology’s growth prospects and finally, increase the number of marine-derived products on the market.

Therefore, the study aims to assess scientific research activity in the field of marine biotechnology, with a focus on pharmaceutical and food applications, by analyzing the most recent bibliography and research projects with high technological maturity and commercial potential according to the *Technology Readiness Level* (TRL) classification adopted by the European Commission [45]. Horizon 2020 (2014–2020) the leading European funding instrument, represents the case study for selecting research-intensive projects that were most relevant to the study’s aims.

In fact, to reach this purpose, the analysis of scientific literature alone is not exhaustive since, although it leads to the identification of innovative solutions under study, it provides no information on the industrial feasibility of a technology. The work’s novelty lies in an innovative approach that combines the analysis of the literature with an evaluation of market trends, defined by high-TLR projects funded in the same sector. This approach is crucial to gaining a complete overview of the techniques under development, ranging from the most embryonic technologies (TRL 1–5) to those ready for marketing (TRL 6–8).

## 2. Results

The results of the study are based on a combination of two investigative methods, respectively geared to the section of emerging technologies and innovative business models with predominant research activity.

Application of the PRISMA method for the systematic analysis of literature enabled a quali-quantitative classification of the most significant advances in scientific research in the field of marine biotechnology through extensive screening of sources. Meanwhile, the application of queries to the H2020 dashboard platform supports the research and statistical analysis of the most relevant and mature projects (TRL 6–8) funded by Horizon 2020 that have led to new goods, products and services being placed on the market. The projects identified are considered case-studies that help determine trends and future industrial research prospects, with a focus on pharmaceutical and food applications.

A detailed explanation of these methodologies is provided in the “Materials and Methods” section.

### 2.1. Systematic Quantitative Analysis of the Literature: Development Trends and Top Players

This section presents the results of the systematic analysis, highlighting the chronological evolution of scientific production over the past ten years. The leading research institutions in the field of marine biotechnology are classified according to their impact factor and area of technological specialization.

Between 2010 and 2020, 620 publications relating to marine biotechnology with applications in the food and pharmaceutical industries were created. A full list of selected articles is available in “Appendix A: articles divided by challenge and technology” a Appendix A. Their time evolution is shown in Figure 1.

The average number of publications per year in the period 2010–2019 was 59, with a minimum of 35 articles published in 2012 and a maximum of 86 in 2018. The number has more than doubled from 2010 (36) to 2019 (70). The year 2020 was not yet finished and was therefore not considered in this statistical analysis.

The 620 articles selected were published in 222 scientific journals or books, including the Marine Drugs guide, with 100 articles and an impact factor of 4.379, mainly dedicated to microalgae (25) and invertebrates (14), to which biochemical characterization and molecular techniques (15) and omics techniques (10) were applied. The top 10 publishers in terms of number of publications, shown in Table 1, assembled almost 50% of the articles, with an average impact factor of 4.278.

The sample of publications under study was classified according to the country of affiliation of the first author and, in the case of multiple countries, the second one was included as well. The continent in which marine biotechnology in the context of pharmaceuticals and food is most widely studied is Asia, with 312 publications (50%), followed by Europe with 208 (33.5%), as can be seen in the graph in Figure 2.

The top ten countries producing scientific research in the field of marine biotechnology applied to the pharmaceutical and food sectors are shown in Table 2 and are responsible for over 70% of scientific production. Iran and the United States are in tenth and eleventh places respectively, with an equal number of publications.

China (102), India (79) and South Korea (61) occupy the top positions in terms of number of publications. China also has the highest impact factor (4.174) in Asia, while for India and South Korea it is below 3. The analyzed studies use predominantly biochemical and molecular characterization techniques, optimization of culture conditions (22.5% of Chinese publications) and recombinant techniques (13%), while in India and Korea it is those concerned with pharmacological analysis that stand out (at 10% and 21% respectively). India is also distinctive in having the largest number of publications on biocatalysis and biosynthesis techniques (8 publications, 10% of Indian publications). Alongside South Korea, it is responsible for publishing a significant number of review articles on the pharmaceutical and food applications of marine biotechnology (respectively 19% and 16% of the publications in the respective countries).

Italy ranks fourth in the world and first in Europe for the production of scientific research. With 55 publications, it is responsible for over a quarter of the total studies published on this continent. The quality of research, expressed in terms of the average impact factor, ranks third in Europe behind Portugal and Spain. The graph in Figure 3 shows the top 5 countries’ main technologies for the number of publications. As in the Asian countries, biochemical and molecular characterization techniques are prevalent in Italy (25.5%); omics techniques (16%) and pharmacological analysis techniques (14.5%) also play an important role. Similarly to Italy, pharmacological analysis techniques (13%) rank third in Portugal in terms of use, while techniques for the optimization (16%) of culture conditions rank second.

Table 3 shows the 10 most productive research centers globally in terms of number of publications.

China, which was the first country to produce scientific research on the pharmaceutical and food applications of marine resources, is home to four of the ten leaders in the field, with an average impact factor of 3.478. The Ocean University of China is in first place with regard to the absolute number of publications (18, i.e., 3% of the total publications). Typical studies have been carried out using biochemical and molecular characterization techniques (39% of the center’s publications) and recombinant techniques (22.2%). In terms of the number of publications, it is followed by the Chinese Academy, which stands out for its recombinant (33% of the publications produced by the center) and screening (27%) techniques. Italy has two research centers of excellence, with an average impact factor of 3.525: the Anton Dohrn Zoological Station is first in Europe and second in the world and specializes in the use of omics technologies (30%), and the National Research Council of Naples, with a production mostly dedicated to biosynthesis and biocatalysis techniques (20%).

The scientific production of the National University of Ireland (6.15) stands out in terms of impact factor, with a wide variety of technologies used ranging from the use of bioinformatics to the optimization of culture conditions.

### 2.2. Systematic Qualitative Literature Review: Resources, Technologies and Challenges of Blue Biotech

The results presented in this section were obtained after a reclassification of the literature sample identified according to the most widely used marine resource, correlated with the challenge under study and the technology used.

Microorganisms, which include bacteria, fungi, protists, and microalgae, are probably the marine resources with the greatest potential for pharmaceutical and food applications, and therefore the focus of 60% of the literature reviewed. The most widely used microorganisms are microalgae (23% of total publications) followed by bacteria (18%), while the least studied are viruses (less than 1%). Figure 4 shows the marine organisms used in the 620 publications analyzed in this study.

Although the most widely used organisms are the microalgae, the main phylum—the taxonomic group above the class—that is used or analyzed in 61 of the publications is the proteobacteria, which belongs to the bacteria kingdom. Other important phyla of the bacteria kingdom used in the publications under analysis are actinobacteria (35 publications) and firmicutes (31 publications). Although the cyanobacteria belong to the bacteria kingdom, they were considered as part of the microalgae category because their features are more similar to the latter. After the phyla ochrophyta (42 publications) and chlorophyta (36 publications), the third most widely used microalgae phylum in marine biotechnology for pharmaceutical and food applications are the cyanobacteria, cited in 27 publications. Figure 5 shows a graphic representation of the different phyla used in the publications analyzed.

The articles selected were divided into 12 categories according to the technology used for the discovery of bioactive organisms and substances or for the large-scale and/or sustainable production of these. The “materials and methods” section in the case of research articles, and the entire text in the case of reviews and book chapters, were taken into account for the classification. In articles based on more than one technology, the most innovative one was chosen or, in the case of traditional technologies only, the main one. Table 4 provides a definition of the 12 technologies identified.

Figure 6 shows a breakdown of the articles according to the technology and their trend from 2010 to 2020.

“Biochemical and molecular characterization”, as a series of methods designed to test a substance or organism’s properties, is the most widely used technology in the publications selected. It is applied in 150 studies, amounting to 24% of the total, with a constant annual growth in production that peaks in 2018 (27, i.e., 31% of the publications produced in 2018 and 18% of the publications that use biochemical and molecular characterization techniques). In second and third place in terms of the number of publications are “optimization of growing conditions” (15%) and “pharmacological analysis” (13%) respectively, followed in fourth place by “omics technologies” (10%), which has also seen a steady growth trend over the last decade. The categories “drug discovery” and “systematic literature review”, used in 11% and 1.1% of the articles in question, respectively, were not included in the above graph and will not be analyzed because they do not refer to a specific laboratory technique.

With reference to the time trend shown in Figure 6, almost all categories show a positive percentage increase in the number of publications in the last five years (2015–2019) compared with the 2010–2014 period except for “chemical synthesis category” that shows a percentage change of—40%. A percentage increase of less than 50% is observed in the following categories: “screening” (+8.3%), “optimization of culture conditions (+38.9%), “biocatalysis” (+40%) and “optimization of harvesting and extraction methods” (+47%). Technologies that have more than doubled the number of publications in the last five years (2015–2019) compared with the 2010–2014 period, are bioinformatics (+100%), pharmacological analysis technologies (+131.8%), and omics technologies (+141%). A substantial increase is also observed for “recombinant DNA technologies” (+55%), followed by “biochemical and molecular characterization” (+73%).

The scientific production selected by this study has been classified according to three categories corresponding to the identified challenges relating to the exploration of the marine environment (353), improvement of the product yield and characteristics (141)—simplified into “productivity” for the graphs and tables—and sustainability (52), in order to assess the evolution of scientific production with focus on the most innovative and promising technologies, goods and products for the full development of marine biotechnology. Excluding the categories “drug discovery” and “systematic literature review”, the breakdown of the publications according to the challenge they face is as follows:Discovery: 353 publicationsProductivity: 141 publicationsSustainability: 52 publications.

The results of the breakdown according to technology are shown in Figure 7.

The discovery of new organisms and substances is dominated by techniques to determine the biochemical and molecular characteristics (40% of the publications assigned to the challenge of discovery), applied to new organisms or substances to understand their properties and potential applications. Pharmacological analysis (21%) and omics technologies (15%) are also significant, gaining several percentage points compared with the total distribution of publications according to technology. The category “screening” only accounts for the 7% of the publications in the discovery challenge but it is important to notice that almost every article in this category (25 publications out of 26) are also assigned to the discovery challenge. On the other hand, technologies used to optimize cultivation conditions are becoming less significant (1%) and are, therefore, less practicable in the discovery phase of blue biotechnology products than those mentioned above.

The main techniques used to increase the productivity of an organism, the yield of a product or to improve its properties fall into the category of “optimization of cultivation conditions”, which is used in more than half of the articles assigned to the productivity challenge (55%).

Recombinant techniques also play a significant role (25%) and are mainly used to genetically modify producer organisms in order to maximize their productivity or produce improved food varieties.

Collectively, the “optimization of cultivation conditions” and “recombinant techniques” categories cover 80% of the technologies used in the publications assigned to this challenge, demonstrating their effectiveness in increasing productivity of the desired product. The techniques of biochemical and molecular characterization, which ranks first in both the general distribution of technologies and the challenge for the discovery of new organisms or substances, only cover 3% in this case. Other techniques that are central for the discovery challenge, such as the pharmacological analysis and the omics technologies, become less significant in this category only accounting for 1% and 4%, respectively.

To ensure sustainable production, techniques to optimize the collection or the extraction of bioactive substances (31% of the publications assigned to the sustainability challenge) and the category of biocatalysis and biosynthesis (23%) seem to be fundamental. A considerable number of publications assigned to the latter category (7 out of 12) deal with the green production of metal nanoparticles. The use of techniques to optimize cultivation conditions (19%) that aim to replace traditional and unsustainable methods is also significant. These three categories account in total for 73% of publications that fall in the sustainability challenge.

### 2.3. Analysis of Projects Using the H2020 Dashboard: The Case-Study

The analysis aims to assess the trend of applied research by analyzing projects in the marine biotechnology sector supported by the main European funding program, with the ultimate aim of identifying the most innovative models among those that have led to the authorization and marketing of a product of marine origin. This approach, combined with the previous literature analysis, enables the definition of the sector’s development prospects.

The study analyses 29 projects funded in the last Horizon 2020 planning period (2014–2020), which were found through the European Commission platform H2020 dashboard and divided into the Blue-Growth funding program—BG (14), Bio-Based Joint Undertaking—BBI-JU (10) and SME- Instrument—SMEINST (5), which supports research and demonstration activities with different part-financing rates (from 58% to 100%). The total funding received was € 149,433,492, out of a total investment of € 172,177,229, i.e., an average of approximately 87%. *Annex 1* lists the projects identified.

The most relevant information used for the subsequent analysis is: type of call, project, start date, duration, TRL range, budget, contribution received from the European Union, the challenge faced and technology used.

As seen in Figure 8, there is a constant annual trend of funded projects in the blue biotechnology sector for pharmaceutical and food applications, peaking at double in 2017 due to the support of the BBI-JU calls. The total funding provided by the Commission for these projects was €149,433,491, divided up as follows:€98,480,377 for the BG projects, with an average funding of €7,034,313€43,868,120 for the BBI projects, with an average funding of €4,386,812€7,084,994 for the SME instrument, with an average funding of €1,416,999

The distribution of projects according to the funding received in relation to their TRL is shown in Figure 9, with three distinguished groups.

The projects funded by the SME instrument are concentrated in the bottom right-hand corner of the graph (Figure 9) and illustrate an investment of less than 2 million euros—with an average of almost one-third of that of the BG projects—and a high TRL (8 or more). Moreover, the uniform diameter of the bubbles corresponding to the projects funded by this instrument shows that, for all these projects, the percentage of funding received in relation to the investment is consistent and amounts to 70%. The other two clearly distinguishable groups are the BBI-JU (TRL 4–5) and BG (TRL 6–7) projects). The technologies and products studied by the BG are slightly superior to the BBI in terms of technological maturity, with a TRL value above 6 for 57% of the BG projects. For the BBI, 70% of the projects selected have a TRL of less than 5. The two types of calls also differ in the funding obtained. 90% of the BBI projects received a Commission contribution of between 3 and 6 million euros, while only 20% of the selected BG projects belong to this category. The remaining 80% received more than 6 million euros in funding.

In order to find the most mature technologies that might overcome the main obstacles to the marketing of biotechnology products, each project was categorized according to challenge and technology, as shown in the graph in Figure 10.

First of all, it should be noted that, to describe the technologies used in the projects, only 6 of the 12 categories identified for the classification of the articles are sufficient. Of these, the most widely used are the optimization of culture conditions and the harvesting and extraction methods, whether in combination or individually. These three categories contribute to the classification of more than 65% of the projects selected. They are particularly relevant if only the projects with the highest TRL (above 6) are considered: 11 out of 17 projects use this type of technology and as seen from the graph in Figure 10, these are particularly promising for the productivity and sustainability challenge.

The omics and screening technologies, used in 2 and 3 projects respectively, are instead specific in making the process of discovering new substances produced by marine organisms to be used for pharmaceutical and food applications more effective. 3 of the 5 projects that fall into these two categories have a TRL of 4.00, and only two projects exceed the value of 6. By contrast, recombinant DNA techniques are used in 4 projects exclusively to increase the productivity of an organism, the yield of a product or to improve its characteristics. 75% of these have a TRL greater than 6 and are therefore good examples of promising technologies for placing on the market.

Of the 29 projects selected, only 16 show a TRL higher than 6, indicating use of a highly mature technology. In general, the technologies used for the challenge of sustainability and increasing productivity (optimization of cultivation conditions and harvesting and extraction methods, recombinant technologies) show a higher TRL than those used for the discovery challenge (screening and omics technologies). Pharmacological analysis techniques are an exception (used in one project only). Although they are used for the discovery of a new product, the project concerned has a high TRL (8). The high TRL projects concluded so far have been selected as business models for a more detailed analysis, making a total of 6, of which 5 are funded by the phase 2 SME instrument and 1 by a Blue Growth call.

Table 5 summarizes the selected projects with the respective technology and the challenge addressed. Budget and funding data can be found in Table 5 above.

## 3. Discussion

Marine biotechnology is an emerging technology aimed at the sustainable exploitation of marine bioresources through their conversion into high value-added products or biofuels [10]. Many products derived from marine organisms are currently being studied for their excellent bioactive properties, and biotechnology is increasingly seen as a sustainable way to exploit their potential. The market data for this sector is also extremely encouraging, and it is estimated that the marine biotechnology market will grow significantly over the coming years, with an incremental growth of 2.5 billion dollars from 2020 to 2024 [46]. In Europe, this value will reach 1301.85 million dollars, with the pharmaceutical and food sectors responsible for over 60% of the added value [15]. The main factor that will drive the marine biotechnology market is clearly the enormous biodiversity of the marine environment, [47] along with an increased consumer appetite for alternative, natural products that are both effective and beneficial to human health [48].

In order to find trends with the highest competitive potential that will dominate the pharmaceutical and food markets in the near future, the study performs a systematic analysis of the literature produced from 2010 to 2020 and identifies projects funded by the main European programs in the last planning period (2014–2020) based on the use of marine biotechnology.

Quantitative analysis of the production of literature on blue biotechnology in the last 10 years shows a substantial increase, as it is ever greater considered a strategic tool in exploiting the potential of marine resources and the fight against the depletion of ecosystems.

From a regulatory point of view, many countries have adopted policy tools to exploit marine ecosystems, partly in response to the action taken by the 1992 Convention on Biological Diversity, which promotes their protection, and the United Nations Sustainable Development Goals, including the conservation of the planet and seas. This combination of tools regulates and steers the market towards sustainable solutions, directing public and private research interests towards marine biotechnology’s pharmaceutical and food applications.

Although China, the leading producer of scientific research, has no strategy specifically dedicated to marine biotechnology, it has been encouraging the marine sector, considered a strategic pillar for economic growth, since 1996 through government subsidies and tax incentives. Marine biotechnology is seen as the main tool to “exploit the sea using science and technology”, one of the Chinese government’s strategic goals. The development of centers of excellence on this territory has encouraged cutting-edge research on the application of marine biotechnology in the pharmaceutical and food sectors. These include the Ocean University of China, which hosts two Key Labs of Marine Biotechnology, Marine Genetics and Breeding and the Chinese UNESCO Centre for Marine Biotechnology, and the Chinese Academy of Sciences, whose Institute of Microbiology has also collaborated in several European projects, including PharmaSea (research into new bioactive compounds) and MGATech (enzymes from hypersaline environments) [49].

India also has a prominent role among the Asian countries. Since 1988, the government Department of Biotechnology has been running an Aquaculture and Marine Biotechnology program to support national and international projects in this sector. This is considered a key element in using bioresources and the development of products that benefit society. The program also includes developing a national marine center, responsible for coordinating research, whose development is still awaiting launch [50].

Among the European countries, Italy generates the most scientific research, and is one of the main contributors to the development of the European Strategy for Blue Growth, which, adopted in 2012, is the first planned contribution of the European Commission to identify long-term, sustainable growth strategies based on aquaculture, renewable energy from the sea, coastal and maritime tourism, mineral resources and marine biotechnology [51].

The recognition of the key role of marine biotechnology is stimulated by the promotion of an Italian Strategy for the Bioeconomy (BIT) and the emergence of the Blue Italian Growth (BIG) Technological Cluster, which have underlined the importance of marine bioresources for the sustainable development of the country [52,53]. Over 30% of Italian publications are generated by the Anthon Dohrn Zoological Station, one of the Italian centers of excellence in the field of marine sciences and marine biotechnology with possible industrial applications, which benefits from multidisciplinary expertise in ecology, physiology, genomics and transcriptomics, biochemistry and cell biology [54].

Despite its considerable potential applications, marine biotechnology is an emerging field, typically associated with dynamic research but with a still limited number of products on the market [55]. Systematic analysis of the literature has identified three macro-areas, or main challenges, respectively associated with the *discovery of new marine organisms* and their bioactive substances, the *increase in productivity*, yield or improvement of the characteristics of a product, and the *sustainable production* of a substance or optimization of the seafood supply chain, especially in relation to waste management. The majority of the selected publication (almost 65%) are assigned to the discovery challenge, meaning that discovering new bioactive molecules is a priority of basic research unlike increasing productivity or product sustainability.

Of the 12 categories of techniques identified, the most used in absolute terms is the “biochemical and molecular characterization” which includes a series of standardized techniques focused on the analysis of organisms, enzymes, or newly discovered molecules before evaluating their possible application. This category also ranks first in the discovery challenge, representing today the most used technology to discover new bioactive molecules. However, as a basic technology, it does not provide the necessary added value to innovate the drug discovery process and overcome some of the limits to marine products’ marketing. Therefore, although a large number of publications using it, biochemical and molecular characterization technology cannot be considered among the emerging trends of future research. Other relevant technologies are optimizing culture conditions for the productivity challenge and optimizing harvesting and extraction methods for the sustainability challenge.

Analyzing the most used technologies in absolute terms in the selected publications gives a useful overview of the basic research carried out so far but is not functional to define the development trends of marine biotechnologies. To this purpose, it is more interesting to analyze their growth during the period considered (2010–2020). In this context, there was an increase in the number of studies dedicated to bioinformatics, omics technologies and pharmacological analysis whose use has more than doubled in the last five years (2015–2019) compared with the previous period. These technologies are mostly used for the discovery of new organisms or products of marine origin, in line with the objectives of basic research generally carried out by universities and research centers.

Pharmacological analysis techniques are dominated by the development of cell growth inhibition and antimicrobial activity tests to assess and quantify the bioactivity of substances and, in some cases, the safety profile. The growing consumer demand for products that are truly effective and also safe for human health is the main driver for the development of this technology.

Omics and bioinformatics technologies are increasingly responding to the limitations presented by marine exploration. Among the most innovative approaches that support the drug discovery process, the analysis of DNA, metabolic and protein profiles of organisms, the creation of metagenomic libraries from samples that cannot be grown in the laboratory [56,57], and the creation of in silico methods such as modelling and molecular docking, [58] are the most popular. Moreover, in several cases, bioinformatics techniques can be combined with the omics techniques to analyze the vast amounts of data generated by this technology, especially in the case of metagenomics [59,60].

The analysis of contributions from the scientific community is supplemented by that dedicated to the main projects in the marine biotechnology sector funded at European level. This has made it possible to identify the most promising business models that have successfully overcome the main obstacles to commercialization and resulted in new products obtained from marine resources being placed on the market.

From 2014 to 2020, the European Commission allocated a total of 149,433,491 euros to demonstration projects in the Blue Growth sector with food and pharma applications, through the Horizon 2020 Programs (Work Program 2—Food Security, Sustainable Agriculture and Forestry, Marine, Maritime and Inland Water Research and the Bioeconomy), the Bio-Based Industries Joint Undertaking and the Phase 2 SME instrument. Considering the subdivision of projects by technology, the reduced number of categories used to describe the projects compared to the publications in literature is evident. It highlights how some up-and-coming experimental technologies used in the laboratory are not marketable and suitable for industrial production.

The projects with a high TRL (above 6), mostly financed by the SME instrument, address industry trends analysis and identify the most promising technologies. These include techniques for optimizing cultivation conditions and techniques for the extraction and collection of products, used individually or in combination in 65% of projects with high technological maturity, to solve sustainability and productivity challenges. The most common high-potential business models include developing new biorefineries or sustainable aquaculture methods for high value-added products, increasing the yield of microalgae, sometimes by means of innovative bioreactors, and those aimed at the production of feed to improve quality and animal welfare. Further examples of promising technologies in applied research are recombinant DNA technologies used to improve the quantity and the quality of fish meat or high added value products extracted from algae. However, these techniques are only applied to projects launched in recent years, highlighting their industrial potential and, at the same time, their innovativeness.

Finally, more innovative recombinant technologies, combined with optimization techniques, are the most promising technologies in applied research and will overcome several challenges to commercialization in relation to both productivity and sustainability.

With reference to marine-derived products, exemplary business models are aimed at nutraceutical production, including extracts of omega-3 polyunsaturated fatty acids and photosynthetic pigments, the main marine-derived products marketed so far. The additional properties of these new products, such as more effective formulation or combination with other compounds that enhance their bioactivity, make them more innovative than products already on the market. However, there are no true pharmaceutical products between the selected business models. This is because although these products are highly profitable, as shown by the high market share of the pharmaceutical sector which, in Europe alone, amounts to just over 35% [15], they require many years and considerable investments before they reach the market, due to the stringent regulation of pharmaceutical products, which discourages small and medium-sized enterprises in particular from taking this path [3].

Therefore, it is expected that the nutraceutical business models will lead the market trend in the marine biotechnology sector in the upcoming years. As clear market-pull examples, functional foods and nutraceutical products are produced in response to customer needs, answering the strong necessity for sustainable solutions to produce food to satisfy the growing food requirements without competing with land crops. Furthermore, these products fit perfectly into the growing trend for green molecules that, at the same time, show a beneficial effect on human health. The methods of cultivation of microalgae, the most abundant sources of nutraceutical products at the industrial level, are characterized by a minimal environmental impact, integrating perfectly in the ecosystem and optimizing the use of resources. As in the Blue Iodine II project, one of the business models analyzed in the study, algal biomass can be exploited through an integrated biorefinery approach with limited energy inputs and recycling waste produced by the fish industry in a circular approach, promoting the sustainability of the entire supply chain.

Alongside nutraceutical applications, the aquaculture sector is also strongly growing as it represents an alternative solution to the unsustainable exploitation of marine fauna. This leads to a growing demand for low environmental impact feeds that also promote animal wellbeing and high production yields.

Food and feed applications in the marine biotechnology sector represent future scenarios and define the most interesting sectors for upcoming investments.

Through the comparison between a qualitative and quantitative analysis of the literature and the market trends, the study does not only represent a simple description of the marine biotechnology sector but also provides several practical applications, representing a starting point for future research aiming at overcoming all those obstacles that limit the marketing of products derived from marine organisms.

The constant dialogue between research and industry is essential for a product developed from research to respond to market demands. This study represents a connection point between research and industry, trying to bridge the gaps between these two environments and promote their collaboration. It provides information to researchers on market trends, identifying the most attractive technologies and products that will have a greater probability of being commercialized and directing research activities towards market-pull business models. At the same time, by highlighting the products that will lead the marine biotech market in the upcoming years, this manuscript offers industries an effective tool for identifying the most promising investments, especially in the food sector as can be seen from the analysis of business models, more detailed in the next paragraph. The six identified business models perfectly respond to market needs aimed at finding new sources of sustainable food, respond to the willingness of the consumer to recognize an economic value to the marine-based product compared to the standard ones commonly present on the market, integrate perfectly in the ecosystem and optimize the use of resources in a circular approach.

The country leaders in the production of scientific research on marine biotechnology adopt specific policies to encourage the marine and maritime sector, harmonise research, allocation of funds, and legislation related to marine origin products. Sustainability policies, especially in the food sector for the search for new food sources that support the growing needs of the population, will further push the market towards adopting alternative solutions, including especially marine biotechnologies. Therefore, the analysis carried out in the manuscript also promotes evidence-informed policymaking, representing the basis of new positioning documents that will lay the foundations for new planning periods and provide indications on the blue economy’s main investment sectors.

### 3.1. Business Models: From Research to Market

The following section describes the six business models identified among the projects analyzed in this publication, defining the purpose and the main results obtained in the project.

#### 3.1.1. Blue Iodine II. Boost BLUE Economy through Market Uptake an Innovative Seaweed Bioextract for IODINE Fortification II

The Blue Iodine II project, funded in 2016 as part of the SME instrument, is developing new economic algae-based products rich in iodine to combat iodine deficiency in three main target groups for which no dedicated products are yet on the market: infants/7- to 14-year-olds, pregnant and breastfeeding women and the elderly.

By developing the best cultivation conditions, propagules can be grown in ground tanks and obtained on a large scale. The proximity of the installations to the sea can make sea water available throughout the year, at minimal pumping costs. Seaweed production near seafood farms in the open sea also makes it possible to exploit the waste from sea bream, rich in nutrients, as food for the algae, which helps avoid discarding waste into the marine environment. The project includes the development of a biorefinery process to exploit algal biomass using cold extraction and filtration techniques to obtain purified extracts.

The product obtained (IODOBEM) is a natural extract rich in iodine and other nutrients such as proteins, vitamins, and minerals, with numerous advantages over products already on the market. Firstly, it has a higher concentration of iodine (30%), and vitamin C (300%). The proteins extracted also contain essential amino acids and stabilize iodine during assimilation. IODOBEM also avoids sodium chloride overdose, a problem often found in synthetic products. Instead, it is rich in iron, which works in synergy with iodine to support thyroid function, and copper, another essential mineral for the body. The specific production mechanism also enables a 10–30% reduction in price of these products compared with their competitors [61].

#### 3.1.2. CryoPlankton 2. Cryopreservation of Marine Planktonic Crustacean Nauplii for Innovative and Cost-Effective Live Feed Diet in Fish Juvenile Aquaculture

The Norwegian company Planktonic AS, responsible for the CryoPlankton2 project, is dedicated to the development and production of live, appropriately preserved feeds for use in aquaculture. In detail, the innovative proprietary technology of this company is concerned with the cryopreservation of crustacean nauplii, reanimated as live individuals after thawing to be used as natural food for fish larvae due to their excellent nutritional properties, which are superior to standard feeds. The process is very simple: the nauplii sampled from the Norwegian coast is frozen in appropriate bags in a cryopreservative presence, which keeps the larva intact. The bags are then placed into liquid nitrogen, where they can be stored almost indefinitely. The larvae are revitalized through a process that takes only one hour a day and requires neither cultivation nor specific feeds for the nauplii [62].

The project, funded by the SME instrument, aims to solve one of the main obstacles to the production of fish in aquaculture, namely the juvenile stage during which live diets are used. This is because the juveniles have a fairly low survival rate at this stage: even for established species such as sea bream and sea bass it rarely exceeds 25%, and for new aquaculture species such as yellowtail and tuna, mortality is even higher. Juvenile fish perform better when fed nauplii that are cryopreserved by the innovative technology developed by Planktonic AS than they do on traditional diets: growth rate increases, mortality rate decreases, deformities are reduced, and the health of the fish larvae improves due to the absence of pathogens in the innovative feed.

The project aimed to set up the production process of this innovative feed to achieve industrial production volumes, improve the vitality of the nauplii after thawing, and decrease the variability between batches. By the end of the project the technology had improved, going from a TRL of 7 to a TRL of 9, with a significant increase in production, reaching 8 million tons. The processes of handling the nauplii between the collection site and the cryopreservation facility and transporting the cryopreserved product to the aquaculture installations were optimized by determining the optimal temperature and density [63].

This technology therefore represents a breakthrough in the breeding of several marine species, both because of the excellent nutritional properties of crustacean nauplii, and due to the simple, inexpensive technology developed for the cryopreservation and subsequent thawing [62].

#### 3.1.3. INMARE. Industrial Application of Marine Enzymes: Innovative Screening and Expression Platform to Discover and Use the Functional Protein Diversity from the Sea

INMARE is a 4-year project launched in April 2015, coordinated by Bangor University (UK) and funded by the Horizon 2020 program. With a budget of 7,396,689.65 euros and almost 6 billion euros of funding, this ambitious project aims to innovate the process of discovering new enzymes by isolating potential candidates from the marine environment more quickly and efficiently.

Through targeted sampling in both previously explored and unknown marine environments, the project has generated one of the largest genomic and metagenomic collections of enzymes, which is useful for the project and future biodiscovery processes. This library includes approximately one thousand useful enzymes, most with characteristics suitable for industrial applications: 94% of these are already available in expression systems and 32% of the enzymes have been fully characterized.

Based on this collection, together with innovative in silico and in vitro screening technology, 15 ready-to-use enzymes have been developed and tested for industrial operations.

The project, which was completed in March 2019, proved an unprecedented success, resulting in more than 60 publications, four patent applications, and a start-up dedicated to the creation of enzymes optimized from natural sources [64,65].

This process demonstrates the potential of the marine environment as a source of new biocatalysts and highlights how omics technologies—particularly metagenomics—combined with the correct high-throughput screening method, represent one of the most efficient strategies for the discovery of new bioactive compounds.

#### 3.1.4. LIFEOMEGA: Innovative Highly Concentrated Omega-3 Specialized Nutrition Product

The purpose of LIFEOMEGA, a project funded by the phase 2 SME instrument in 2017, was the industrial development and subsequent commercialization of a nutritional product with a high eicosapentaenoic acid (EPA) content to improve the well-being of cancer patients undergoing chemotherapy.

The innovation of the product developed by the project lies in its unique, patented formulation. It is actually an emulsion that is easily administered in other liquids, making it easier to swallow than pills and capsules. The flavored product improves patient compliance, and its high concentration means that 3 grammes of EPA can be taken per day in a single 20 mL dose. Pharmacological studies carried out on the product have also shown that the specific type of product formulation increases the bioavailability of EPA in the body, and that the emulsion shows potent anti-inflammatory effects.

The product is designed to boost the health of cancer patients by improving treatment outcome, facilitating recovery, shortening hospital stays, and improving patients’ quality of life during and after treatment.

LIFEOMEGA is currently being tested in other clinical trials to determine the level of nutritional improvement in patients and the biological activity of the product [66,67].

#### 3.1.5. SMILE: Slimming and Memory-Booster MIcroaLgae Extract

The SMILE—Slimming and Memory-Booster MIcroalgae Exctract project, funded by the SME instrument in 2016, aims to develop a nutraceutical product derived from microalgae with proven benefits for weight control and the support of cognitive function, two of the main challenges facing modern society. The effectiveness of the carotenoid fucoxanthin in these two objectives had already been showed in previous studies. However, the microalgae used in its production have certain disadvantages, such as micropollutants and sustainability problems.

The French start-up Microphyt has developed an innovative, patented technology based on special 5000-litre tubular photobioreactors, enabling it to grow several microalgae of particular interest that are difficult to cultivate through traditional technology, with a total absence of external contamination and total control over the growing conditions. The green extraction process helps maintain the sustainability of the entire production.

The active ingredients in SMILE are fucoxanthin and omega-3 fatty acids, which act synergistically at different levels of brain function. They are dissolved in a matrix based on natural coconut oil together with an antioxidant, also of natural origin, which protects the active molecules from oxidation and extends the shelf life of the product.

During the project, Microphyt carried out pharmacological efficacy and safety studies in the laboratory and preclinical studies on mice, confirming the efficacy of the active molecule pool. The application for authorization of these ingredients was submitted to the European Commission at the end of 2018.

Two patents have also been registered. The first is concerned with microalgae cultivation: managing the availability of light inside the photobioreactor increases the accumulation of pigments up to two times the data obtained in the literature. The second, on the other hand, concerns the use of SMILE ingredients and their composition to prevent cognitive disorders in humans and animals [68,69].

#### 3.1.6. VOPSA 2.0: Value Omega 3 and Astaxanthin Products from SeaAlgae

VOPSA 2.0 is a 2-year project funded through the SME instrument at the end of 2016. The project was developed by the Spanish companies Neoalgae and Bicosome. It aims to sustainably meet the growing demand for omega-3, an essential oil needed for correct functioning of the body that also has cardioprotective and beauty-enhancing properties, and astaxanthin, a carotenoid with a powerful antioxidant effect. These compounds are normally obtained from oily fish such as salmon in the case of omega-3, or from oceanic krill or by chemical synthesis in astaxanthin. Overexploitation of these organisms combined with chemical processing makes the production of these substances unsustainable and polluting.

Neoalgae has therefore developed a system for the production of omega-3 and astaxanthin using microalgae (particularly the species *Nannochloropsis gaditana, Isochyrsis galbana* and *Haematococcus pluvialis*), microorganisms that are easy to cultivate and can be used with no negative impact on the marine ecosystem. The cultivation system used provides two separate areas for the production of the two different compounds, with different nutritional requirements and growth conditions; both areas include columnar photobioreactors, controlled raceways—or flow-through systems—and tubular photobioreactors.

In addition to this sustainable cultivation system, there is also a supercritical extraction method in which carbon dioxide is applied to the freeze-dried biomass. This avoids the hazardous use of traditional organic solvents, further contributing to the sustainability of the system. This approach, carried out after pre-treatment that increases the final yield, has facilitated the separation of highly pure omega-3 and astaxanthin compounds that are free of pollutants, have no after-taste and are suitable for vegans. It should be emphasized that the whole process has been achieved at competitive and stable production costs: the final price per liter for the production of omega-3s is €2.5, and €3.5 for astaxanthin.

In vitro and in vivo tests on microalgae products have confirmed their safety and efficacy for dermatological use [70,71].

Neoalgae has used these products to develop a line of cosmetic items known as Alskin. Their flagship product is a face cream containing astaxanthin-rich *Haematococcus pluvialis* extracts. Astaxanthin oil (extracted from *Haematococcus pluvialis*) and omega-3 oil (from *Nannochloropsis*, *Isochrysis* and *Phaeodactylum*) and food supplements based on spirulina and astaxanthin have also been produced.

Bicosome, on the other hand, has developed an exclusive, patented technology for releasing microalgae on the skin. Through this system, the active components of the cream can penetrate deep into the dermal layers of the skin, resulting in superior antioxidant, anti-inflammatory and protective effects. These products will be marketed under the name Bioalgae^®^ Xanthin and Bicoalgae^®^ omega-3 for the treatment of skin disorders such as acne, atopic disorders and ageing [72].

The wealth of products derived from this project demonstrates this technology’s effectiveness in generating marketable products while ensuring sustainable production.

## 4. Materials and Methods

The data collection process used in this publication followed two independent yet rigorous, precise steps to obtain replicable and reliable results.

The literature available on the main repositories was analyzed using the PRISMA method, a recognized, systematic methodology that provided an assessment of the state of research in blue biotechnology for pharmaceutical and food applications.

Once the projects funded under H2020 were identified as case studies, the H2020 dashboard platform was then invaluable in gathering information on the main projects in the sector of interest, focusing on those with a high TRL (Technology Readiness Level). The purpose of this second level of investigation was to identify the most advanced products and technologies with a high probability of reaching the market, in order to identify research trends and some use cases of particular interest.

### 4.1. Literature Review: The PRISMA Method

The PRISMA method (Preferred Reporting Items for Systematic Reviews and Meta-Analysis), chosen for the analysis of the literature referenced in this publication, is a systematic, explicit, replicable methodology for the identification, selection and evaluation of articles relevant to the research topic and the collection of data from the studies included in the review. The process, carefully stated in the guidelines, follows a four-step diagram that uses a funnel system to help researchers optimize the reporting system and select relevant publications for study [73]. The steps carried out are shown schematically in Figure 11. This type of approach therefore minimizes bias, thus providing reliable results from which to draw conclusions [74].

In order to diversify the publications found as much as possible, three different databases were selected for the initial search: Pubmed, Scopus and ScienceDirect. Scopus is the most comprehensive citation and abstract database which, with over 25,000 titles in both journals and books, provides a comprehensive view of world research, not only in science, technology and medicine, but also in the humanities, arts and social sciences [75]. On the other hand, ScienceDirect is dedicated to scientific and medical research. It therefore constitutes a less extensive but equally valid repository, enabling access to 2500 scientific journals, many of which are open access, and 39,000 books [76]. Finally, the selection of contributions in food and pharmaceutical fields is supported by Pubmed. This database includes more than 30 million citations, primarily in the medical and related fields such as biological sciences, behavioral sciences, and biomedical engineering [77]. The use of repositories that allow access to articles on marine biotechnology with different but comparable criteria has therefore led to the collection of a broad spectrum of sources in a uniform, easily replicable manner.

The keywords used as search terms were a combination of the terms “drugs”, “food”, “pharmaceutical application” or “nutraceutical application” with the term “marine biotechnology” or the variant “blue biotechnology”. The search focused on the literature produced (articles, reviews, book chapters) from 2010 to April 2020, identified by the occurrence of keywords in the title or abstract, and keywords chosen by the authors of the publication or used by the system to catalogue articles in the database.

Following an initial screening, 1883 publications were identified, including 481 in Pubmed, 1232 in Scopus and 170 in ScienceDirect, as shown in Table 6.

The articles selected were screened according to the funnel approach recommended by the PRISMA method.

The removal of duplicates, i.e., identical articles taken from different databases or with different search terms, led to a total number of 1258 articles, which were subjected to further selection as a result of

The next step consisted of a first-level of screening based on a rapid reading of the title and abstract of the articles. This highlighted 560 non-relevant publications that were mainly concerned with applications other than those of the research topic or made no reference to the marine environment. Some were also eliminated because they referred to another topic altogether: for example, the term “blue” led to the identification of a number of articles that referred to the use methylene blue or other blue reagents for experimental use but were actually focused on a totally different field and could not be considered relevant to the objective of this analysis. However, the search term “blue” still had to be included, because it enabled the selection of significant articles that would never have been found using the variant “marine”.

The 702 articles obtained from these first steps were then read more thoroughly, and a further 78 publications were discarded. There were a number of reasons for this elimination, but they were mainly related to the articles being too general in terms of the source, with no recognizable precise focus on the marine environment, or a lack of application for the organisms described and analyzed in the article. Albeit to a lesser extent, other articles were discarded, in spite of the initial filter, for the following reasons: the text was not written in English; the main application on which the article focused was neither pharmaceutical nor food, or was not biotechnological; the type of publication was different from those selected; the publication, especially in the case of book chapters, was a simple, very general presentation of marine biotechnologies and was not useful for understanding the progress of the research. Table 7 shows the number of articles discarded for each reason.

The funnel screening process resulted in the selection of 620 publications for inclusion in a qualitative and quantitative analysis.

More specifically, from a quantitative point of view, we chose to make a temporal assessment, dividing the articles according to the year of publication, and a geographical assessment, analyzing the origin of the first author. We also decided to rank the journals according to which had published the most on the topic in question.

On the other hand, the qualitative analysis highlighted, for each article selected, the organization used to obtain a given product or to which the application was addressed, and the analysis or production technique used. The publications were also divided into three categories according to the problems faced by the sector that the stated technologies and products were aiming to resolve: the discovery of new products, increasing yields and large-volume production, and sustainability.

Taken as a whole, this data gives a detailed picture of the state of research in marine biotechnology for pharmaceutical and food applications, making it possible to identify both established and innovative technologies to overcome the main challenges faced by the sector.

### 4.2. The European Case-Study: Horizon 2020 and Related Funding Measures

With a financial allocation of 80 billion euros, Horizon 2020 (2014–2020) is the largest European research and innovation program, with an investment line dedicated to “Blue Growth” [78]. This is supported by both the SME instrument, converted into the Enhanced European Innovation Council (EIC) pilot in June 2019, and the Bio-based Industries Joint Undertaking funding program which forms an excellent funding opportunity for projects in the marine biotechnology sector. The first is an instrument part of the H2020 program dedicated to funding projects with high technological maturity (starting TRL greater than 6), which are aimed at the commercialization of innovative products, services and business models [79]; the second is instead a public-private partnership between the European Union and the Bio-based Industries Consortium, aimed at the development of new biorefineries that sustainably transform natural resources into bio-based products, materials and fuels [80].

The authors of this study selected the European funding programs as a case study from which to select projects funded in the sector of interest, to determine the research applied in the pharmaceutical and food sector of marine biotechnologies. In fact, although not representative of research carried out throughout the world, due to the obvious geographical limitations, the data from these projects is the only data contained in a single, easily accessible, investigable database. Therefore, the data collection and project selection are statistically reliable as they are derived from a primary source whose data has not been reprocessed. Compared with other financing measures operating on a global scale, the European case study enables selection of a greater number of projects, as it refers to an entire continent rather than an individual country or group of countries.

Finally, the projects identified were subjected to further selection aimed at identifying business models, with a high TRL, which were representative of the most promising technologies and products, i.e., with a higher probability of reaching the market.

#### 4.2.1. Use of the Platform: Filtering and Data Extraction Criteria

The H2020 dashboard [81] is an interactive platform that works on a series of tabs, enabling the collection and filtration of data on Horizon 2020, thus facilitating data sharing and access to information on the main European funding program. Thanks to the “H2020 funded projects” functionality, this platform provides detailed data on projects funded, not only by H2020 but also BBI, and returns a list of projects matching the filtering criteria used. For this reason, the authors of this study selected the H2020 dashboard as the most suitable platform to identify the main projects funded in the field of marine biotechnology for pharmaceutical and food applications [82].

In order to identify the largest number of relevant projects, three analyses were carried out in the same rigorous manner, one for each type of call previously identified as critical to the funding of marine biotechnology projects.

The tab selected was “H2020 projects”, since it provides a list of desired projects downloadable in Excel format with the following information: (1) title of project; (2) acronym of project; (3) thematic priority; (4) Number of participants in project; (5) Contribution received from program; (6) ToA (Type of Action); (7) Net contribution received by program; (8) project number; (9) link to CORDIS; (10) Topic code; (11) Description of topic. Moreover, in all three cases, the filter “Thematic priority” was used to select “Biotechnology” and “Food security, sustainable agriculture and forestry, marine and maritime and inland water research” because they are related to marine biotechnology; the other topics were not considered relevant to the analysis.

A second filter “call ID” was then applied to select the projects funded under a specific call identified by a unique code. Different calls were selected according to the case:Projects funded under the “Blue Growth” calls in all 3 WPs: 11 calls were selected with initial code H2020-BG (H2020-BG–2014-1; H2020-BG-2014-2; H2020-BG-2015-1; H2020-BG-2015-2; H2020-BG-2016-1; H2020-BG-2016-2; H2020-BG-2017-1; H2020-BG-2018-1; H2020-BG-2018-2; H2020-BG-2019-1; H2020-BG-2019-2) making a total of 55 projects.Projects funded under the BBI: 7 calls were selected, including 3 with initial code H2020—BBI—PPP (H2020-BBI-PPP-2014-1; H2020-BBI-PPP-2015-1-2; H2020-BBI-PPP-2015-2-1) and 4 with code H2020-BBI-JTI (H2020-BBI-JTI-2016; H2020-BBI-JTI-2017; H2020-BBI-JTI-2018; H2020-BBI-JTI-2019) making a total of 203 projects.Projects funded through a phase 2 SME Instrument: 3 calls were selected with initial code H2020-SMEINST (H2020-SMEINST-2-2014; H2020-SMEINST-2-2015; H2020—SMEINST-2-2016–2017) making a total of 97 projects. In this case, it was decided not to consider phase 1 projects, since this funding is reserved for assessing the technological and commercial feasibility of a product or technology.

#### 4.2.2. Selection of Relevant Projects

The projects contained in each Excel file found on the Horizon 2020 platform by applying the filters described above were subjected to further manual screening, to select those relating to blue biotechnologies with applications in the food and pharmaceutical sectors. The data available on CORDIS, the Union’s public database and primary source of information on projects financed by Europe, were essential for analyzing the purpose, topics and technologies of each project and then selecting the most relevant. 29 projects were identified during the screening. These were divided as follows:Projects funded under the Blue Growth calls: 14Projects funded under the BBI: 10Projects funded through a phase 2 SME Instrument: 5

#### 4.2.3. Search for Additional Information for the Selected Projects and TRL Assignment

The information contained in the dataset of the 31 selected projects was supported by data relating to the budget, the project duration, the coordinator and the type of organization used or addressed by the application, based on the information contained in CORDIS. As in the case of the articles, the projects were likewise classified according to the challenge of the sector they aim to solve.

An essential piece of data added was the TRL, i.e., the indicative technology maturity level reached by each project. The ToA provided by the dashboard combined with the specifications of the call to which the project applied was central in this respect. The approximate correlation between TRL and ToA used for the classification is as follows:Research and Innovation Action (RIA): TRL < 5Innovation Action (IA): 6 < TRL < 7SME Instrument phase 2: TRL = 8

Statistical analysis was conducted on all previously selected projects, irrespective of the TRL. However, since the objective of this survey was to identify the most promising technologies and products with the highest probability of reaching the market, the business models were identified among the 17 projects with a TRL greater than 6. Of particular interest were 6 completed projects chosen as business models for which information on the technology used and products obtained is already available.

### 4.3. Limitations of the Study

Despite the careful procedure followed while collecting data from the literature and the H2020 platform, the study has some limitations that must be taken into account when drawing conclusions.

#### 4.3.1. Limitations of the Bibliographic Research

The number of publications found using the PRISMA method should be considered limited, for three reasons:the use of a finite number of keywords. The final number of articles selected could be further increased by diversifying the keywords used.all stages of the screening of articles were conducted by a physical operator. It is therefore possible that some errors of assessment were made. However, out of the total number of articles found, any oversights should not statistically affect the result.only a 10-year period was considered (2010–2020). Increasing the time frame would further increase the number of articles. However, a time frame of 10 years is more than sufficient for the purposes of the publication in question, Publications for the year 2020, as it is not yet over, must also be considered incomplete. Therefore, this last year was not considered when assessing the time trend of the publications, as it could be misleading.

For the analysis of technologies used in the publications identified it must also be considered that the category “drug discovery”, while being relevant, since it is used in 11% of the articles, together with the “Systematic literature review”, were not included because they do not refer to a specific laboratory technique and are therefore not applicable to this study, which aims to identify innovative experimental technologies. Excluding these two categories, the number of publications on which the technology analysis was performed was 546.

#### 4.3.2. Limitations of the H2020 Dashboard

The European case-study is not representative of applied research worldwide due to the obvious geographical limitation. Furthermore, the projects selected do not represent all technologies and products funded at European level, for the following reasons:only H2020, SME and BBI programs were considered. Although these have very high budgets and are the most widely used for the funding of marine and maritime projects, the possibility that other smaller programs could support relevant projects cannot be overlooked. However, due to the absence of a platform that would enable a rigorous analysis like the one carried out on the H2020 dashboard, and due to the much lower funding, these funding sources were not considered in the study.Only a small number of calls are considered. Given the cross-cutting nature of marine biotechnology, it must be remembered that projects in this sector could also find a source of funding in other calls, under other thematic priorities (e.g., “Climate actions, environment, resource efficiency and raw materials” or “Secure, clean and efficient energy”)

## 5. Conclusions

Marine biotechnology in pharmaceuticals and food applications is an emerging sector that is globally encouraged by an increasing number of policy and financial instruments. Through the analysis of 620 publications and 29 projects, this study identifies the most promising technologies and business models aimed at the identification of new substances, increasing the yield of those already known and ensuring sustainable production.

Omics, pharmacological analysis, and bioinformatics technologies drive the trends in scientific research and are considered fundamental tools for the discovery of new substances and organisms as candidates for industrial applications.

Techniques for the optimization of culture conditions, harvesting and extraction methods, combined with recombinant techniques, are central to most industrial models with immediate commercial exploitation, especially in food and nutraceutical applications. The emergence of this sector has been facilitated by the growing consumer demand for healthy foods that also have beneficial effects on health, combined with a less stringent legislation than that which applies to pharmaceutical substances.

Although this review takes account of some limitations, including the subjectivity of the authors in classifying publications and the limited number of projects analyzed, which only refer to the European continent, it not only provides an updated qualitative and quantitative analysis of the literature produced with reference to applications in pharmaceuticals and food, but also highlights the trends in basic and applied research in the sector. It thus promotes evidence-informed policymaking and represents a starting point for future research aimed at overcoming all those obstacles that restrict the marketing of products derived from marine organisms.

## Figures and Tables

**Figure 1 marinedrugs-19-00061-f001:**
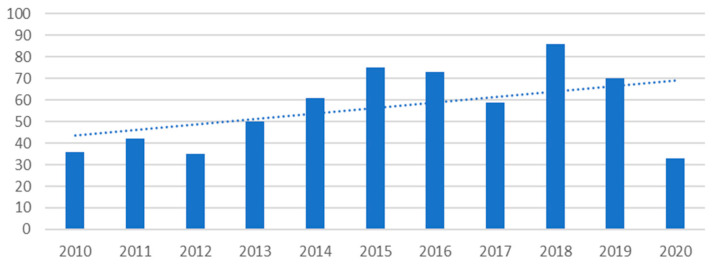
Worldwide scientific production from 2010 to 2020.

**Figure 2 marinedrugs-19-00061-f002:**
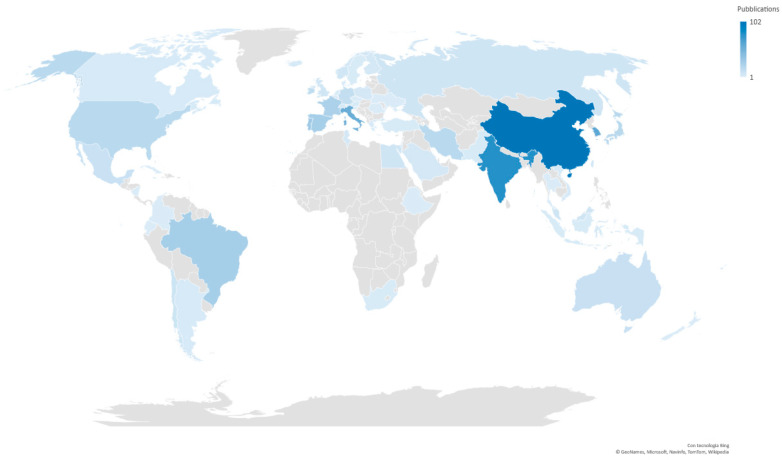
Geographical distribution of publications in the world.

**Figure 3 marinedrugs-19-00061-f003:**
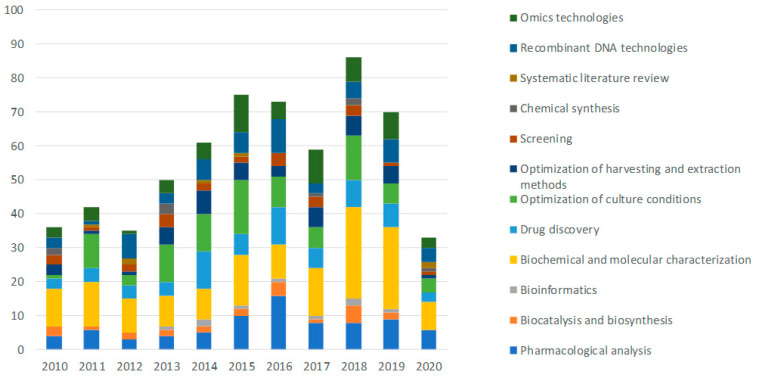
Breakdown of publications by country and the technologies they use. Only the top 5 countries producing scientific research in the area under analysis were considered.

**Figure 4 marinedrugs-19-00061-f004:**
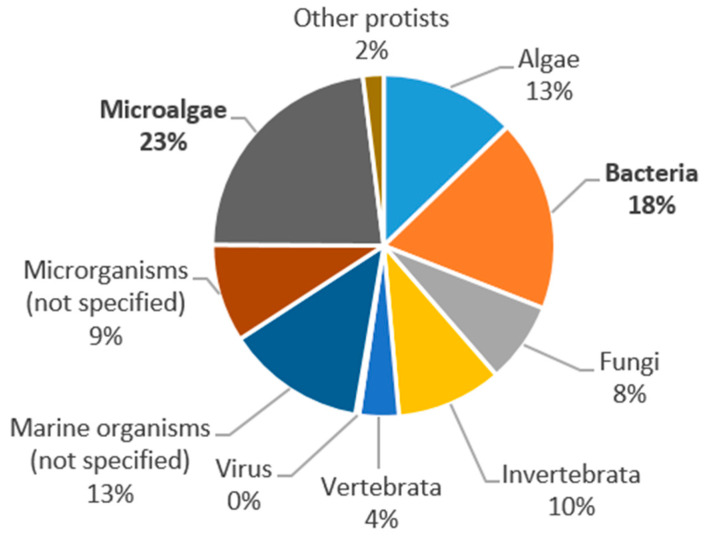
Distribution of marine organisms used in pharmaceutical and food applications of marine biotechnology, by publication.

**Figure 5 marinedrugs-19-00061-f005:**
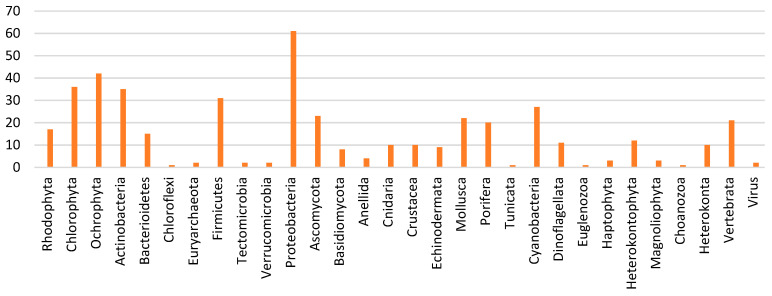
Marine organisms used in pharmaceutical and food applications of marine biotechnology, subdivided by phyla. Publications that mention many organisms, with no specific focus on one or some of them, were not included in the total.

**Figure 6 marinedrugs-19-00061-f006:**
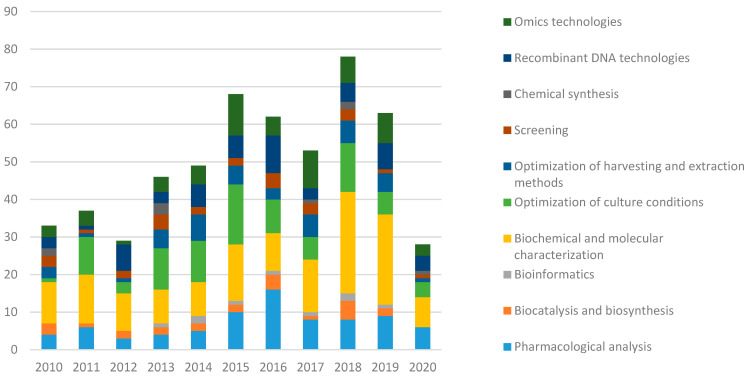
Breakdown of articles according to technology and their time trend during the period 2010–2020.

**Figure 7 marinedrugs-19-00061-f007:**
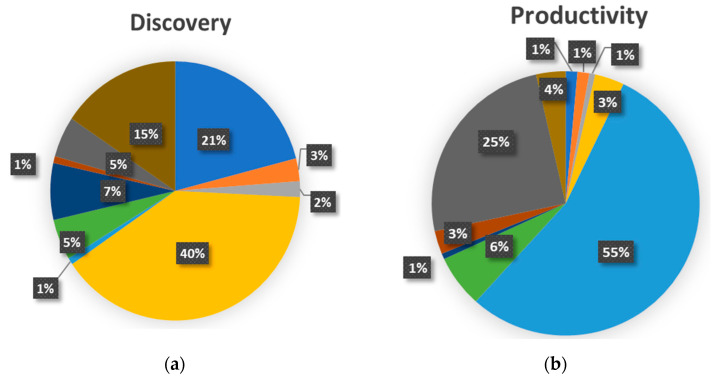
Distribution of articles according to technology broken down by challenge: (**a**) Discovery; (**b**) Increased productivity; (**c**) Sustainable production; the key at point (**c**) refers to all three graphs.

**Figure 8 marinedrugs-19-00061-f008:**
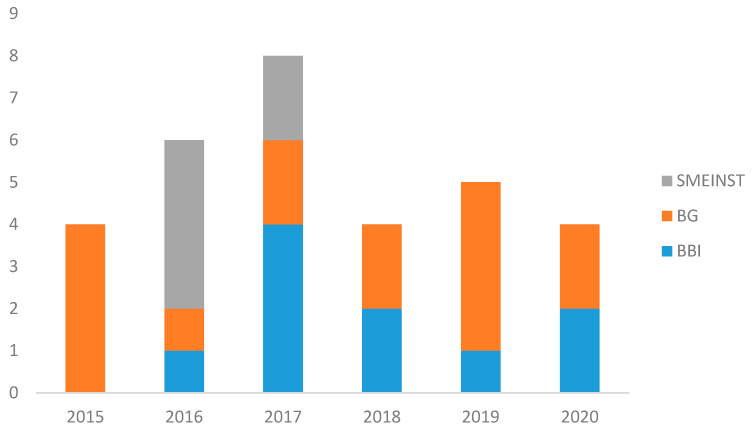
Time trend of projects funded during the 2014–2020 period.

**Figure 9 marinedrugs-19-00061-f009:**
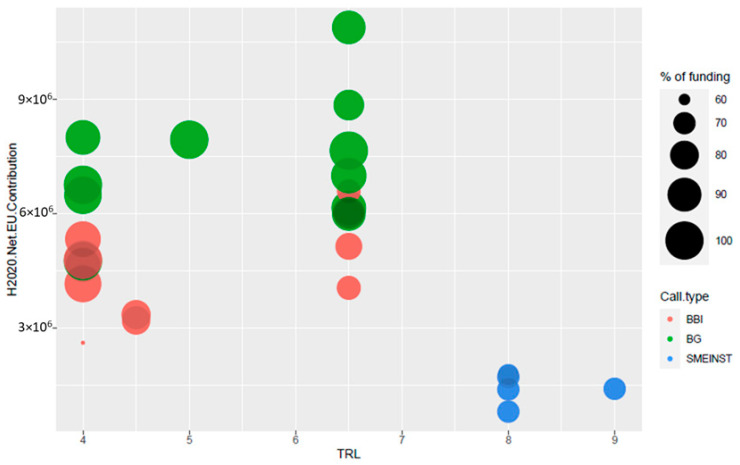
The figure shows the relationship between funding received and TRL for each project. The projects are shown in different colors depending on the type of call (BBI in pink, BG in green, SMEINST in blue), while the size of the bubbles indicates the percentage of funding in relation to the investment made. The graph was constructed using the R program.

**Figure 10 marinedrugs-19-00061-f010:**
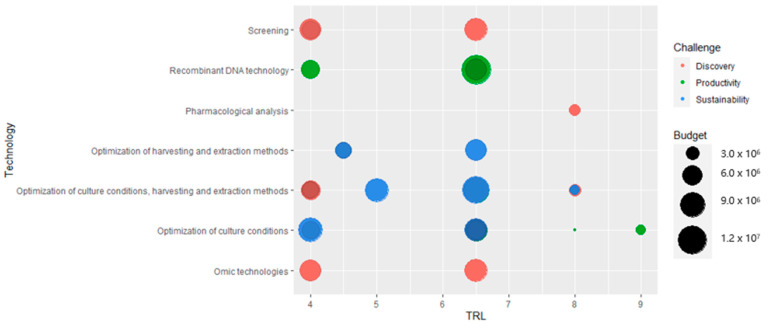
The figure shows the relationship between TRL (*x*-axis), technology used (*y*-axis) and total investment dedicated to the project (bubble size). The projects are shown in different colors according to the challenge faced. The graph was constructed using the R program.

**Figure 11 marinedrugs-19-00061-f011:**
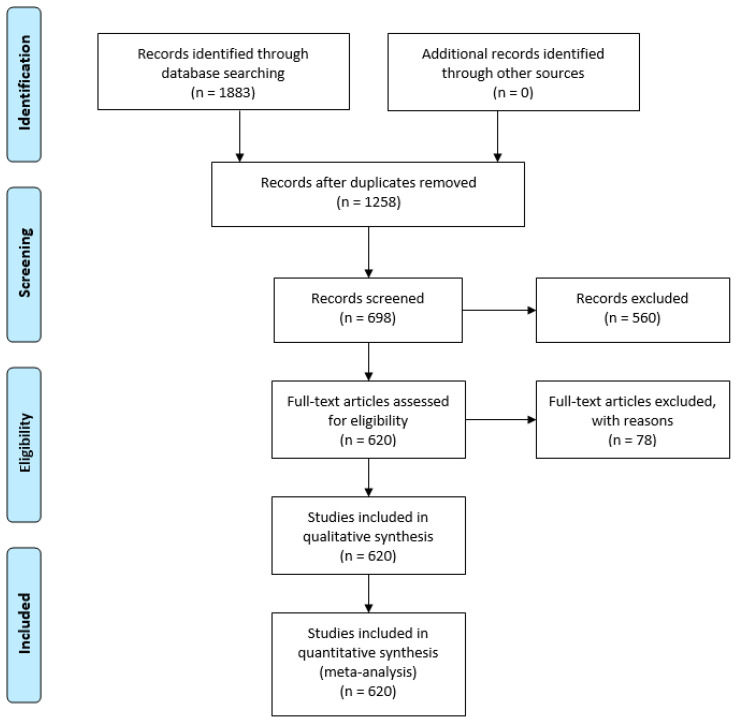
PRISMA Flow Diagram. Schematic view of the diagram followed when applying the PRISMA method with the number of items selected for each step [73].

**Table 1 marinedrugs-19-00061-t001:** Top 10 journals that have published articles on the applications of marine biotechnology in the food or pharmaceutical sectors.

Journals	Publications	Impact Factor
Marine Drugs	100	4.073
Applied Microbiology and Biotechnology	34	3.530
Journal of Microbiology and Biotechnology	33	1.975
Marine Biotechnology	28	1.200
Bioresource Technology	23	7.539
Biotechnology Advances	15	10.744
Biotechnology and Bioprocess Engineering	12	2.213
Journal of Bioscience and Bioengineering	12	2.366
International Journal of Biological Macromolecules	10	5.162
Algal Research	9	4.008

**Table 2 marinedrugs-19-00061-t002:** Top 10 scientific research-producing countries in the world.

Country	Publications	Medium Impact Factor
China	102	4.174
India	79	2.703
South Korea	61	2.658
Italy	55	3.530
Portugal	31	4.926
Spain	26	5.329
Brazil	26	3.286
France	24	2.143
Japan	18	3.153
Iran	17	2.694
USA	17	3.429

**Table 3 marinedrugs-19-00061-t003:** Top 10 scientific research organizations in the world.

Nation	Research Centers	Publications	Medium Impact Factor
China	Ocean University of China	18	4.607
Italy	Anton Dohrn Zoological Station	17	3.769
China	Chinese Academy of Sciences	15	3.795
South Korea	Pukyong National University	12	2.963
Italy	National Research Council of Naples	10	3.281
Iran	University of Hormozgan	8	4.23
France	Ifremer	7	3.045
Ireland	National University of Ireland	7	6.15
China	East China University of Science and Technology	6	2.675
China	Zhejiang University	5	2.834

**Table 4 marinedrugs-19-00061-t004:** Detailed definition of identified categories.

Technology	Description
Omics technologies	Use of technologies such as genomics, metabolomics, transcriptomics and proteomics.
Biochemical and molecular characterization	Series of techniques of various kinds that serve to characterize the organism and/or substance produced. Classification also applied to those reviews in which the technology is not specified but organisms or substances are analyzed at the biochemical and/or molecular level.
Recombinant techniques	Techniques for modification of the producing organism, such as genetic, protein and metabolic engineering. It also includes the heterologous production of proteins and metabolites.
Pharmacological analysis	Bioactivity determination of substances, their toxicity and pharmacokinetic properties. By extension, it also applies to substances not for human application, such as drugs or probiotics for aquaculture and bioactive substances for use in terrestrial crops
Screening	Techniques for selecting substances, enzymes, or organisms according to their activity using HTS or conventional screening tests
Optimization of culture conditions	Changes in soil, light source, temperature, and other variables that can affect the growth of the organism. The implementation of bioreactors and co-culture methods comes under this category
Drug discovery	General category that includes those reviews that do not address a specific technology, but provide only a broad, general description of the sector
Chemical synthesis	Synthesis of a substance that is equivalent or comparable with the natural one, or modifying it to increase or vary its bioactivity
Collection/extraction optimization	Techniques that improve the harvesting of the organism cultivated or the extraction of bioactive substances
Biocatalysis and biosynthesis	Catalysis of chemical reactions enhanced by enzymes and chemical synthesis processes carried out by organisms.
Bioinformatics	Bioinformatics techniques for the analysis of organisms and substances
Systematic literature review	Review of the literature or patents that use a systematic method

**Table 5 marinedrugs-19-00061-t005:** Selected case studies.

Call Type	Project Acronym	Challenge	Technology
SMEINST	Blue Iodine II	Productivity	Optimization of culture conditions
SMEINST	CryoPlankton2	Productivity	Optimization of culture conditions
BG	INMARE	Discovery	Omic technologies
SMEINST	LIFEOMEGA	Discovery	Pharmacological analysis
SMEINST	SMILE	Discovery	Optimization of culture conditions, harvesting and extraction methods
SMEINST	VOPSA2.0	Sustainability	Optimization of culture conditions, harvesting and extraction methods

**Table 6 marinedrugs-19-00061-t006:** Articles initially selected for each search term from each database.

	Scopus	ScienceDirect	Pubmed
Nutraceutical application blue biotechnology	4	1	4
Nutraceutical application marine biotechnology	22	5	22
Pharmaceutical application blue biotechnology	18	8	22
Pharmaceutical application marine biotechnology	91	27	63
Drugs marine biotechnology	571	24	120
Food marine biotechnology	287	62	171
Drugs blue biotechnology	170	7	38
Food blue biotechnology	69	36	41

**Table 7 marinedrugs-19-00061-t007:** Articles discarded in the second screening phase for each reason.

Criteria for Exclusion of Articles (Full Text)	Number of Articles Excluded
Language of the text not in English	4
Very general article, not specifically focused on the marine environment	25
Type of publication different from those chosen	2
Application not concerned with pharmaceuticals or food	17
Characteristics/analysis of marine organisms but no mention of their application	19
Non-biotechnological application	4
Introductory article on marine biotechnology, not useful for understanding the state of the art of the research	7

## Data Availability

With reference to the statistical analysis of the most recent literature, the data presented in this study are available in Appendix A: articles divided by challenge and technology. With reference to the analysis of European projects, publicly available datasets were analyzed in this study. This data can be found here: https://webgate.ec.europa.eu/dashboard/sense/app/93297a69-09fd-4ef5-889f-b83c4e21d33e/sheet/a879124b-bfc3-493f-93a9-34f0e7fba124/state/analysis. Selected project are available in Appendix B
Table A1: List of projects selected through the H2020 dashboard.

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
