# Peer review of "Marine Biotechnology: Challenges and Development Market Trends for the Enhancement of Biotic Resources in Industrial Pharmaceutical and Food Applications. A Statistical Analysis of Scientific Literature and Business Models"

_marinedrugs, 2021, doi:10.3390/md19020061_

Round 1

Reviewer 1 Report

The review is well written and useful. I would appreciate comparison of the first and second 5 years within the 10-year period to see the trend of changes but it was not intended and the manuscript is interesting enough in the present form.

Minor remarks:

Figures 3 and 6. As the colors in the legend are not extremely easy to distinguish, wouldn’t it be better to organize the legend in order of appearance of the technologies on the plot, like in Figure 8?

Line 668: “[58]are”, please separate.

Some references lack doi, please complete whenever possible.

Author Response

Dear reviewer,

Thank you for your precious comments on our manuscript.

Please see the attachment to check the point-by-point response to your comments.

Best regards,

Sara Daniotti & Ilaria Re

Reviewer 2 Report

The study presents the research results on ‘Marine Biotechnology: Challenges and Development Market Trends………’. The originality of this manuscript should be emphasized more and this manuscript cannot be acceptable in its present form. Please consider the following comments and suggestion for further revision.

  1. A concise abstract is required. Also, more specific descriptions of authors' finding should be added in the abstract rather than overall result of study. The abstract should state briefly the purpose of the research, the principal results and major conclusions.
  2. There are many research results about marine biotechnology related to market trends. In the introduction section, write the novelty of the work and the problem statement clearly.
  3. The explanation of results and discussion are not enough in Fig 6 and 7. Authors need to provide more detailed explanations and discussions.
  4. In Discussion section give detailed information only citing reference is not sufficient. Formulas should be corrected throughout the manuscript.
  5. Provide the practical applications and future research perspectives of this work before conclusions.

Author Response

(The authors gave the same response as above.)
